

# Carbonate content and stable isotopic composition of aerosol carbon in the Canadian High Arctic

Petr Vodička[1,2], Kimitaka Kawamura[2], Bhagawati Kunwar[2,3], Lin Huang[4] , Dhananjay K. Deshmukh[2,a], Md. Mozammel Haque[2,5], Sangeeta Sharma[4], Leonard Barrie[6]

[1] Institute of Chemical Process Fundamentals, Czech Academy of Sciences, 165 00 Prague 6, Czech Republic

[2] Chubu Institute for Advanced Studies, Chubu University, 1200 Matsumoto-cho, Kasugai 487–8501, Japan

[3] Institute for Space-Earth Environmental Research, Nagoya University, Nagoya 464-8601, Japan

[4] Environment and Climate Change Canada, Science and Technology Branch, 4905 Dufferin St., Toronto, Canada

[5] School of Ecology and Applied Meteorology, NUIST, Nanjing, 210044, China

[6] Atmospheric and Oceanic Sciences Department, McGill University, Montreal, Canada

[a] Now at: Commission for Air Quality Management in National Capital Region and Adjoining Areas, New Delhi 110001, India

*Correspondence to:* Petr Vodička (vodicka@icpf.cas.cz) and Kimitaka Kawamura (kkawamura@isc.chubu.ac.jp)

**Abstract.** The carbon cycle in the Arctic atmosphere is important in understanding abrupt climate changes occurring in this region. Two-years of measurements (summer 2016 - spring 2018) of carbonaceous aerosols at the High Arctic station Alert, Canada, showed that in addition to organic carbon (OC) and elemental carbon (EC), carbonate carbon (CC) was episodically but not negligibly present. The relative abundances of CC in total carbon (TC) ranged from 0-65 % with an average of approximately 11 % over the entire period. Also there was a strong correlation of CC with aerosol $Ca^{2+}$ which is associated mostly with soil dust and to a lesser extent sea salt aerosol. Based on this and the analysis of air mass back trajectories (AMBT), we infer two possible sources of CC in the Arctic total suspended particles (TSP). The major one is the erosion and resuspension of limestone sediments, particularly in the semi-desert areas of the northern Canadian Arctic. Another potential more minor source of CC is from marine aerosol sources including calcified marine phytoplankton shells (coccoliths) introduced into the atmosphere via sea-to air emission.

The CC content significantly influenced the stable carbon isotopic composition ($\delta^{13}C$) of TC. The higher the CC content, the higher the $\delta^{13}C$ values, which is consistent with the strong $^{13}C$ enrichment in carbonates. Therefore, carbonates in Arctic TSP must be taken into account not only in isotopic studies using $\delta^{13}C$ analyses but also when assessing the impact of carbonaceous aerosols on the Arctic climate.





34

## 1 Introduction

The Arctic is a dynamically changing region that is significantly affected by climate change (England et al., 2021). Aerosols are one of the factors influencing the climate, but their effects are subject to significant uncertainties (Carslaw et al., 2013). The uncertainties in radiative forcing is primarily associated with carbonaceous aerosols, most of which is composed of organic carbon (OC). In contrast, a smaller fraction consists of elemental carbon (EC), which is equivalent to optically determined black carbon (BC) (Petzold et al., 2013).

Organic aerosols, i.e., OC in the atmosphere, generally have a cooling effect on the climate (Stjern et al., 2016) except for the part called brown carbon (Laskin et al., 2015). On the other hand, EC or BC has the warming effect, both in the atmosphere (Liu et al., 2020) and on snow surface (Flanner et al., 2007) especially important in the Arctic. In addition, EC and BC measurements are also used to determine the mass absorption cross section (MAC), a fundamental input to radiative transfer models (Mbengue et al., 2021). The MAC is season- and station- specific (Savadkoohi et al., 2024), making it one of the parameters in affecting the influence of aerosols on climate. If either EC or BC is determined inaccurately, the MAC factor will be subsequently biased as well (Chen et al., 2021). Therefore, detailed knowledge of the composition of carbonaceous aerosol in the Arctic is crucial for improving our understanding of their impacts on the climate changes in this region.

Recent atmospheric studies from Tajikistan (Chen et al., 2021) and Tibet (Hu et al., 2023) indicate a significant contribution of carbonates in total suspended particles (TSP), which may have a significant effect on the determination of OC and EC (or BC). Those areas are characterized as arid desert regions with sparse vegetation, large amounts of unconsolidated sediments, and lack of soil moisture, where the wind erosion plays an important role in the aeolian processes such as atmospheric transports and dust deposits. Arctic regions are affected by long range transport of dust and also contributed locally (Groot Zwaaftink et al., 2016; Sharma et al., 2019; Sirois and Barrie, 1999). They have a desert, semi-desert or arid character in some cases (Pushkareva et al., 2016). Recently high-latitude dust sources have been described as a significant climate and environmental factor (Kawai et al., 2023; Kawamura et al., 1996; Meinander et al., 2022). High Arctic semi-desert aerosols have been described as potentially important reservoirs of soil organic matter (Muller et al., 2022), however, the influence of carbonates on the atmosphere in these regions has not yet been systematically studied. Investigation of the ion balance of November to May fresh snowfall at Arctic site Alert over a three year period (Toom-Sauntry and Barrie, 2002) led to the conclusion that missing carbonate (especially in November and May) is the most likely cause of the ion imbalance.





In recent years, carbon in aerosols has also been subject to analyses of the stable isotopic composition ($\delta^{13}$C) as a method for studying atmospheric processes (Gensch et al., 2014; Huang et al., 2006). Several studies within the Arctic have also employed $\delta^{13}$C analysis to study carbonaceous aerosols. Specifically, at the Canadian Alert station, studies investigating $\delta^{13}$C changes in the EC (Rodríguez et al., 2020; Winiger et al., 2019) and springtime $\delta^{13}$C changes in the TC (Narukawa et al., 2008) have recently been published.

Carbonates are strongly enriched in $^{13}$C, with relatively positive $\delta^{13}$C around zero, and can thus affect $\delta^{13}$C values in TC of TSP aerosols, as demonstrated by Chen et al. (2016) for Asian desert dust. In the Arctic region, observed higher aerosol $\delta^{13}$C values are often attributed to dissolved particulate organic matter from marine aerosol sources while the influence of carbonates are ignored (Gu et al., 2023). We hypothesized that the influence of carbonates on Arctic aerosols is not negligible. In this study, we present two years of carbonaceous aerosol observations at the Alert focusing on carbonate content and the isotopic composition of $\delta^{13}$C of TC.

## 2 Experimental

### 2.1 Measurement site and sampling

TSP samples were collected at the WMO Global Atmosphere Watch Observatory at Alert, Nunavut, Canada (82°27'03.0" N, 62°30'26.0" W, 210 m ASL). The Alert site represents a remote Arctic area located on the northeastern tip of Ellesmere Island, which is 817 km from the North Pole (**Fig. 1**). The site has been used for research on atmospheric aerosols since the mid-1980s (Barrie and Barrie, 1990). In terms of carbonaceous aerosols, BC has been measured at the station for decades (Sharma et al., 2004, 2017). Rodríguez et al. (2020) later then reported EC/OC results from TSP between March 2014 to June 2015 with a focus on EC analyses.

For this study, TSP samples were collected from June 13, 2016 to April 16, 2018, using a high-volume sampler on pre-baked (450 °C, 12 h) quartz fiber filters (20 x 25 cm, PALL, 2500QAT-UP). During this period, a total of 93 samples were collected at weekly intervals. The sampled filters were placed in clean glass jars with Teflon-lined caps and stored in a freezer before chemical analyses.



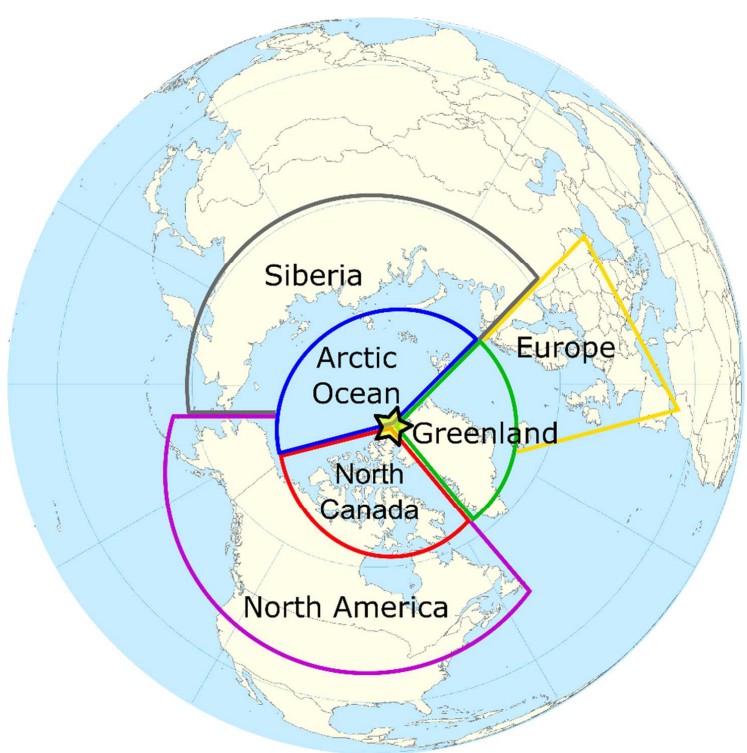

93

**Figure 1.** Map of the site position (asterisk) and geographical boundaries for dominant source regions of backward air mass trajectories (Arctic Ocean - blue, Greenland - green, North Canada islands - red, North America - purple, Siberia - grey, and Europe - yellow). Background map by Wikimedia Commons / Public Domain.

98

**2.2 Analyses**

To obtain the TSP mass concentration, each filter was weighed before and after sampling, and resulting concentrations were corrected for the corresponding field blank filter. Subsequently, we analyzed the samples by two methods to measure the carbonaceous components, with the outputs shown in **Fig. 2**.



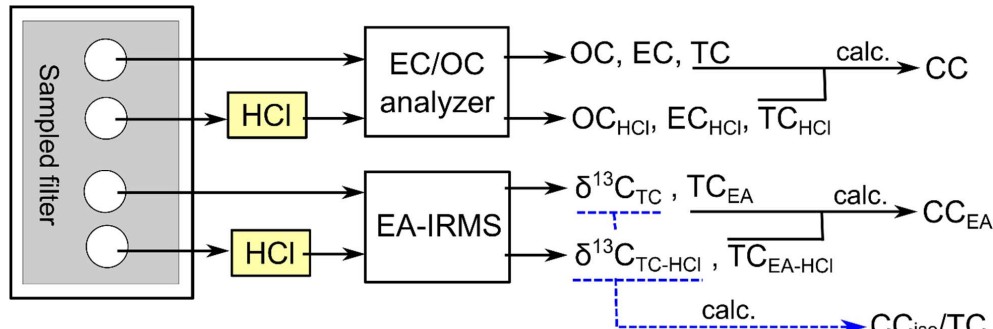

103

**Figure 2.** Diagram showing the method for the measurements of carbonaceous components in quartz fiber
filter samples.

106

Contents of OC, EC and total carbon (TC) were determined using a Sunset semi-continuous analyzer
(Sunset Laboratory Inc., Tigard, OR, USA; (Bauer et al., 2009)) operated in off-line mode. Samples with a
diameter of 16 mm (area 2.01 $cm^2$) were analyzed by Improve_A temperature protocol: step [gas]
temperature (°C)/duration (s): helium (He) 140/120, He 280/120, He 480/120, He 580/120, He-O2 (Ox.)
580/120, He-Ox. 740/120, He-Ox. 840/210 (Chow et al., 2007). Split point between OC and EC was
determined based on laser beam (660 nm) transmittance measurements through the filter during analysis
and raw data were subsequently evaluated by RTCalc726 software (Sunset Lab). The same EC/OC analysis
was performed after exposing the aerosol filters to HCl vapors overnight in a desiccator. From the difference
of TC and $TC_{HCl}$, the content of CC was calculated, which is discussed further in subsection 2.3.

The same samples were further analyzed for the stable carbon isotopic composition ($\delta^{13}C$) of TC by the
method described in more detail elsewhere (Vodička et al., 2022). Briefly, filter cuts (2.01 $cm^2$) were placed
in tin cups, inserted into the elemental analyzer (EA, Flash 2000) and heated to 1000 °C in a helium
atmosphere. At this temperature, carbonaceous compounds are evolved and catalytically oxidized to $CO_2$,
which was isolated on a packed gas chromatograph, and then measured for TC by a thermal conductivity
detector, and finally transferred into the isotope ratio mass spectrometer (IRMS; Delta V, Thermo Fischer
Scientific) via a Conflo IV interface for the analyses of $^{13}C/^{12}C$ ratios. An external standard, acetanilide
(supplied by Thermo Electron Corp.), having a $\delta^{13}C$ of −27.26 ‰ compared to Vienna Pee Dee Belemnite
(VPDB), was used to obtain calibration curves for total carbon (TC) and its isotopic values. Subsequently,
we determined the $\delta^{13}C$ of TC using **Eq. (1)** with relation to the international standard VPDB.

$$\delta^{13}C\ (‰) = [(^{13}C/^{12}C)_{sample} / (^{13}C/^{12}C)_{standard} - 1] \times 10^3 \qquad (1)$$



In this manner, $\delta^{13}C_{TC}$, corresponding to delta TC values on the filter, and $\delta^{13}C_{TC\text{-}HCl}$, representing delta
values on filters after exposure to HCl vapor, were analyzed. The standard deviation of $\delta^{13}C$ measurements
based on triplicate sample analysis was 0.03 ‰.
Through these analyses, we obtained TC from two independent analytical techniques. We observed good
agreement between TC value measured by the EC/OC analyzer and those by EA (r = 0.99). We also
obtained a good agreement even after HCl treatment of the filters (r = 0.98) (**Fig. S1**).

**2.3 Characteristics of carbonate carbon (CC)**
Concentrations of CC, one of the key variables of this study, were calculated from the difference of TC
before and after HCl fumigation (**Eq. (2)**)
$CC = TC - TC_{HCl}$        (2)
**Eq. (2)** defines CC, or nominal CC. **Fig. 3** shows thermograms from OC-EC analyses of a selected sample
without HCl treatment (purple curve) and after HCl treatment (green curve). As shown in **Fig. 3**, the largest
material loss can be seen at the temperature step EC2, but for other samples we observed the largest material
losses in the EC1 and OC4 regions as well. These are temperature steps during which we should expect the
release of carbonate carbon (CC) (Cavalli et al., 2010; Chow et al., 1993). On the other hand, removal of
carboxylic acids (e.g. acetic or oxalic acid) can be expected at temperature steps OC2 and OC3 (Hasegawa,
2022). We also analyzed several carbonate and oxalate salt standards as a control (**Fig. S2**). The thermogram
of the sample with a pronounced peak in the EC2 region (**Fig. 3**) was most similar to that of $CaCO_3$ (**Fig.**
**S2**). Hence, it can be assumed that the significant peaks, removed by HCl in the EC2 region of thermogram,
are carbonate in origin. Nevertheless, it cannot be excluded that some of the prominent CC peaks in the
OC4 region are also from oxalates. The nominal CC may contain other minor carbon components. Here,
however, it should also be noted that the reported values of oxalates in the Arctic (see, e.g., Feltracco et al.
(2021), Svalbard) are an order of magnitude lower than the CC values we analyzed.
It is worth noting here that if we did not analyze CC, it would be determined as either OC or EC based on
the thermogram and automatic determination of the split point (**Fig. 3**). For both EC and OC, we calculated
the percentage of mass removed by HCl fumes as CC using an **Eqs. (3)** and **(4)**.
$EC_{removed}$ (%) = $(EC - EC_{HCl})/EC *100$     (3)
$OC_{removed}$ (%) = $(OC - OC_{HCl})/OC *100$   (4)





In this way, we found that the CC contribution was, on average, 25 % (ranging from 0 to 81%) of EC and
12 % (ranging from 0 to 46 %) of OC which have been inaccurately determined if we had not assessed the
CC contribution. On a relative basis, EC concentrations were more biased in all seasons (**Fig. S3**).
Analyses for CC were performed by two independent methods (EC/OC and EA instruments, **Fig. 2**), and
the resulting CC values show good agreement (r = 0.87, y = 0.95 x, **Fig. S4**). Unless otherwise stated, CC
values calculated from EC/OC analyses are discussed in this study.

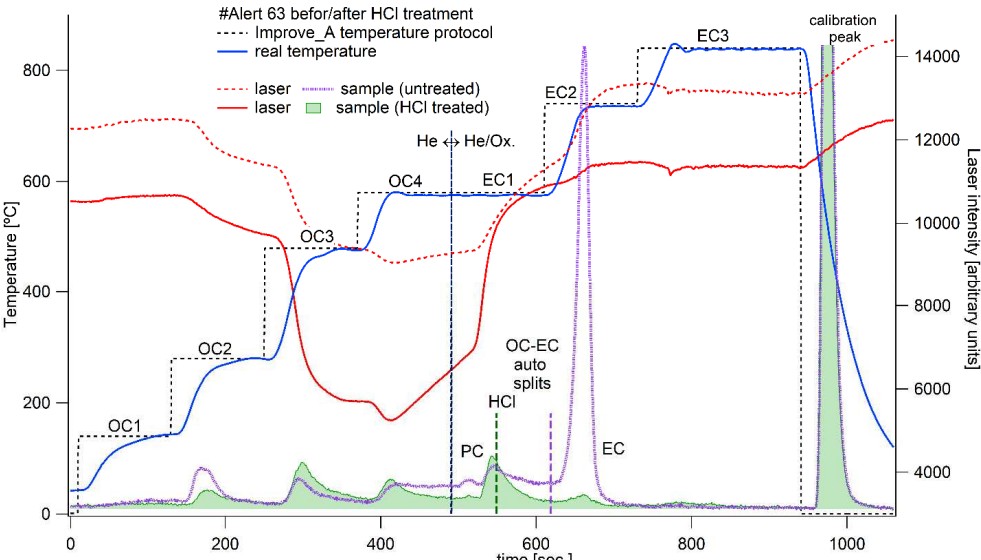


**Figure 3.** Example of EC/OC analysis of untreated (purple thermogram) and treated sample with HCl
fumigation (green thermogram). Sample #Alert-63 was collected from 15-22 May 2017.

**2.4 Estimate of δ$^{13}$C values of CC**
The δ$^{13}$C analyses of TC and TC$_{HCl}$ allowed us to estimate δ$^{13}$C of CC. Here we calculate the δ$^{13}$C values
of released CC on HCl fumigation using the following reaction and the isotopic mass balance equation (**Eq.**
**5**), being similar with the calculation in Kawamura and Watanabe (2004).
TC + HCl $\rightarrow$ CO$_2$ (carbonate carbon) + TC$_{HCl}$



$\delta^{13}C_{TC} = f * \delta^{13}C_{CC} + (1\text{-}f) * \delta^{13}C_{TC\text{-}HCl}$         (5)
, where $f$ means a fraction of CC in TC. From **Eq. 5**, we derived the formula for calculating $\delta^{13}C_{CC}$ (**Eq. 6**).
$\delta^{13}C_{CC} = (\delta^{13}C_{TC} - (1\text{-}f) * \delta^{13}C_{TC\text{-}HCl}) / f$         (6)
The $\delta^{13}C_{CC}$ values were reasonable for a CC content in the TC of approximately above 20 % (**Fig. S5**).
When $f$ (CC contribution) is high, $\delta^{13}C_{CC}$ values are close to zero, supporting that CC is mainly composed
of carbonate, such as $CaCO_3$. However, when the f values are low, the $\delta^{13}C_{CC}$ are highly scattered, indicating
that the released (removed) carbon by HCl is not only carbonate carbon but also contains various types of
carbon including semi-volatile organic acids or unknown species. In the case of organic acids, $\delta^{13}C$ values
can be as high as $-10\text{‰}$ (e.g., Wang and Kawamura, 2006) or positive due to unknown isotope fractionation
during analytical processing. Highly scattered values may also be due to potential analytical errors in EA-
IRMS measurements when $f$ is sufficiently small. If the CC contribution were zero, **Eq. 6** would lead to
division by zero. This may also be the cause of bias and scattering of $\delta^{13}C_{CC}$ values at low CC contributions
(**Fig. S5**).

**2.5 Carbonate estimation from isotopic composition**
We used the isotopic mass balance between $\delta^{13}C_{TC}$ and $\delta^{13}C_{TC\text{-}HCl}$ as an alternative and probably more
accurate method to determine carbonate content ($CC_{iso}$) in TC. The $CC_{iso}$/TC ($f_{iso}$) calculation is based on
the assumption that the $\delta^{13}C$ of carbonates is around 0‰ with an approximate range +5 ‰ to -5 ‰, and no
other compounds are present in this range. We used **Eq.5**, where $\delta^{13}C_{TC}$ and $\delta^{13}C_{TC\text{-}HCl}$ are known and $\delta^{13}C_{CC}$
are given by three different values, covering an approximate range of carbonates (+5 ‰, 0 ‰, -5 ‰).
While $\delta^{13}C_{CC} = 0$, $f = 1 - (\delta^{13}C_{TC} / \delta^{13}C_{TC\text{-}HCl})$     (7)
While $\delta^{13}C_{CC} = +5$, or -5, $f = (\delta^{13}C_{TC} - \delta^{13}C_{TC\text{-}HCl}) / (\delta^{13}C_{CC} - \delta^{13}C_{TC\text{-}HCl})$   (8)
The $f_{iso}$ value is then an average calculated from the three $f$ values obtained from **Eqs. 7** and **8**.

**2.6 Auxiliary data**
Air mass back trajectories (AMBT) were calculated using the National Oceanic and Atmospheric
Administration (NOAA) HYSPLIT model (Stein et al., 2015) at 500 m a.g.l. using a run time 168 h in
GDAS  (Global Data Assimilation System) with 0.5 degree resolution for each sampling days. For
subsequent analyses, we divided the air masses into six sectors as depicted in **Fig 1**.





Meteorological data at 5 min resolution for temperature (T), wind speed (WS) and wind direction (WD)
were provided by Environment and Climate Change Canada. Only WDs between 14 November 2016 and
16 January 2017 were obtained from the NOAA website (https://psl.noaa.gov/arctic/observatories/alert/).
For the purpose of this study, complementary mean values of T and WS were calculated to each sample.
The WD and WS data were used to create wind roses by Zefir software (Petit et al., 2017) and used in
combination with the AMBT data to determine the predominant aerosol origin for each sample. All wind
roses and AMBT by HYSPLIT are shown in supplementary data.
The water soluble ions ($Ca^{2+}$, $Mg^{2+}$, $Na^+$) were measured using an ion chromatography (761 Compact IC,
Metrohm, Switzerland). For this purpose, the filtered samples were twice extracted with 10 ml of ultrapure
water using an ultrasonic bath for 15 min and the aqueous extracts were filtered using a disc filter (Millex-
GV, 0.22 µm, Millipore).

## 213 3 Results and discussion

### 214 3.1 Carbonaceous aerosol composition

Time series of TC, $OC_{HCl}$, $EC_{HCl}$ and CC mass concentrations are shown in **Fig. 4**. An overview of results
is provided for the seasonal variations from 2016 to 2018 in **Table 1** and **Fig. S6** in supplementary material.
An average TC concentration over the entire measurement period was $0.219 \pm 0.147$ µg m$^{-3}$ (median 0.186
µg m$^{-3}$; deviation due to one sample with a concentration of 1.22 µg m$^{-3}$, **Fig. 4a**). Average TC contribution
in aerosol mass ranged from 5 to 14% (**Table 1**). Seasonally, the highest mass concentrations of TC, $OC_{HCl}$
and $EC_{HCl}$ were found in winter and/or spring (**Fig. S6**, **Table 1**).
The fact that $EC_{HCl}$ corresponds to realistic elemental carbon concentrations is demonstrated by comparison
with the study of Rodríguez et al. (2020), who analyzed EC in TSP during 2014-2015. They reported
seasonal EC concentrations that are similar to our observed $EC_{HCl}$, while their CC was part of the OC due
to the use of a different temperature protocol (EnCan-Total-900) during the OC/EC analysis. However,
Rodríguez et al. (2020) did not quantify the contribution of CC to TC.
Concentrations of $OC_{HCl}$ dominate in all seasons (**Figs. 4b** and **5**) but present seasonally different
correlations with ambient temperature. Especially in summer, we observe a significant positive correlation
(r = 0.73) between $OC_{HCl}$ concentrations and temperature (**Fig. S7a**). A similar relationship was also
observed, e.g., at the subarctic station Pallas (Friman et al., 2023), suggesting a biogenic origin of
summertime organic aerosols in Arctic areas (Moschos et al., 2022). In contrast, we observe negative
correlations between $OC_{HCl}$ and ambient temperature in winter (r = -0.15, insignificant) and spring (r = -





0.43). A year-round similar trend for $EC_{HCl}$ (r = -0.39, **Fig. S7b**), supporting previous studies that highlight,
in particular, the anthropogenic contributions to Arctic aerosol during the polar night (Moschos et al., 2022).
However, for CC, we observe no significant dependence on temperature (r = 0.09, **Fig. S7c**).

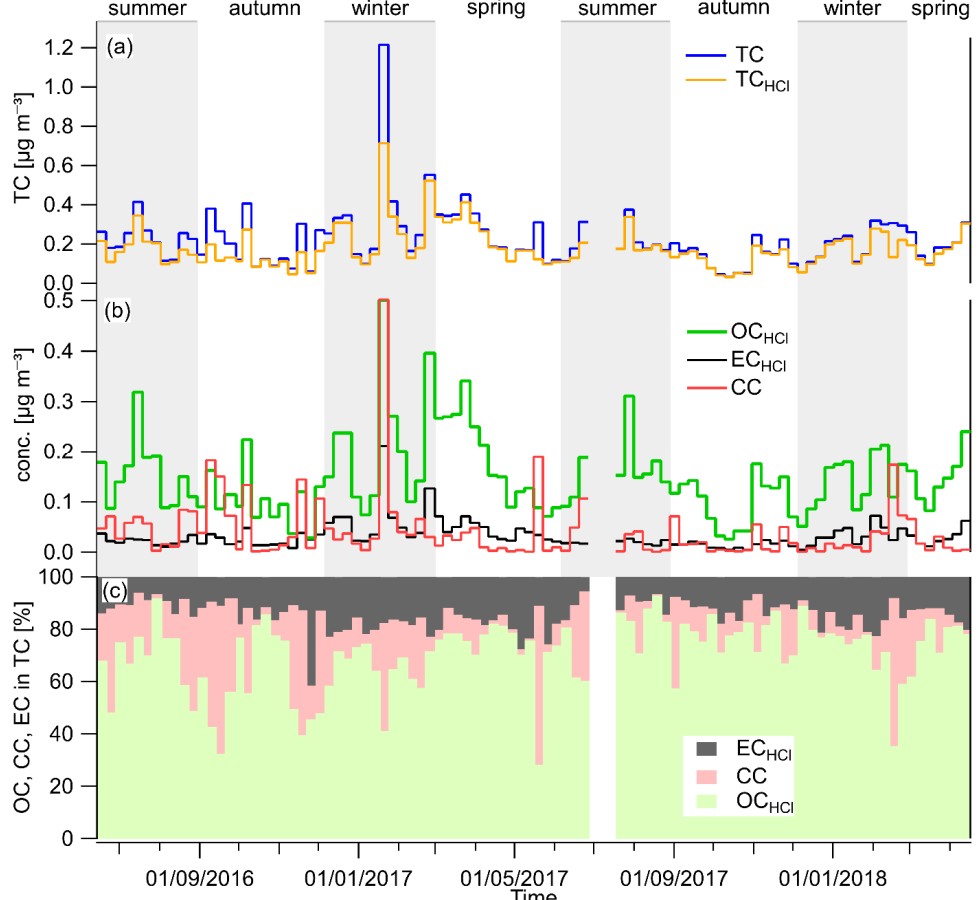

**Figure 4.** Time series of (a) TC mass concentrations before and after HCl treatment of samples, (b) OC,
EC and CC mass concentrations, and (c) their relative contributions to TC in the Alert TSP aerosols.

Time series of TC before and after HCl fumigation (**Fig. 4a**) show that the amount of carbon removed in
the form of CC is neither negligible nor large. However, both mass concentrations of CC (**Fig. 4b**) and its




relative contributions (**Fig. 4c**) show that the CC is often larger than the $EC_{HCl}$ contribution. Specifically,
CC concentrations were highest during both autumn of 2016 and summer of 2016 and 2017 (**Fig. S6c**),
which was reflected also in the relative contributions to TC (**Table1, Fig. 5**).
It is notable to understand the origin of CC. During summer, the contribution of biogenic aerosols (a
potential source of oxalates) is highest, while the landscape is least covered by snow, making the situation
favorable for resuspension of soils eroded from rocks including carbonates. A potential source of carbonates
may come directly from the Canadian Arctic land region, where limestone sediments are reported to be
abundantly present (Groot Zwaaftink et al., 2016; Not and Hillaire-Marcel, 2012; Phillips and Grantz, 2001).
Another likely source of carbonate is marine aerosols as marine organisms contribute carbonate to the sea
(Stein et al., 1994). In the context of wind directions and the effect on $\delta^{13}C$, the origin of CC is further
discussed in the following subsection 3.2.

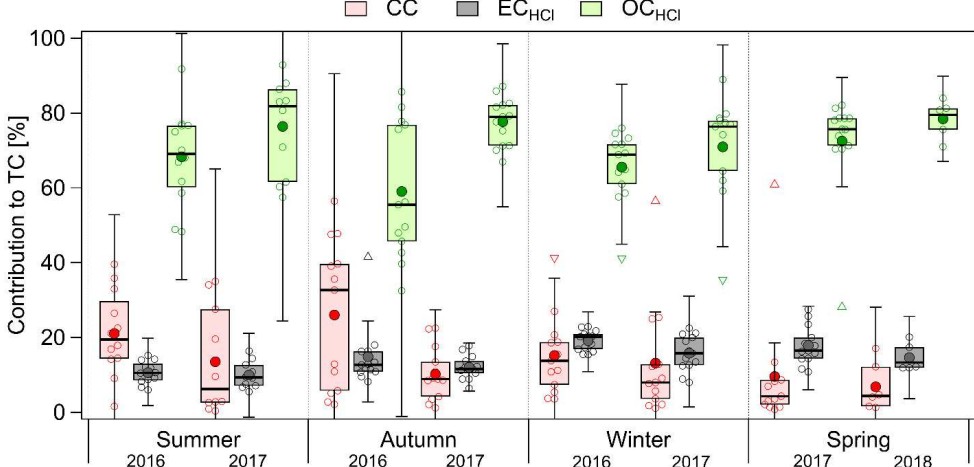


**Figure 5.** Seasonal contributions of CC, $EC_{HCl}$ and $OC_{HCl}$ to TC. The boxes correspond to the interquartile

range (IQR; 25 and 75 percentile) with median represented by the inner solid line. The whiskers correspond

to inner fences range (1.5*IQR), triangles are outliers and mean is represented by large filled circle.


**3.2. Stable carbon isotopic composition**
Removal of CC by HCl fumigation has a significant effect on the measurements of $\delta^{13}C$ isotopic values of
TC (**Figs. 6** and **7**). **Fig. 6** shows a seasonally resolved $\delta^{13}C$ values for HCl-treated (red) and untreated
(grey) samples. We observed significant changes in all seasons except in spring of 2017 and 2018. Here it



is interesting to mention a link with Narukawa et al. (2008), who reported changes in δ¹³C of TC values for
HCl-treated samples between winter and spring at Alert site. Narukawa et al. (2008) show significantly
higher δ¹³C values in spring than winter and related this to the possible contribution of carboxylic acids
(especially oxalic acid). During our observations, the differences in average values of δ¹³C of TC (HCl
treated vs. untreated) were also significant (see red boxes in **Fig. 6** and **Table 1**). Winter and spring δ¹³C
values after HCl treatment in this study (**Table 1**) are similar to those presented by Narukawa et al. (2008)
for the year 2000. Thus, this study confirms that this is a long-term phenomenon likely to occur annually.

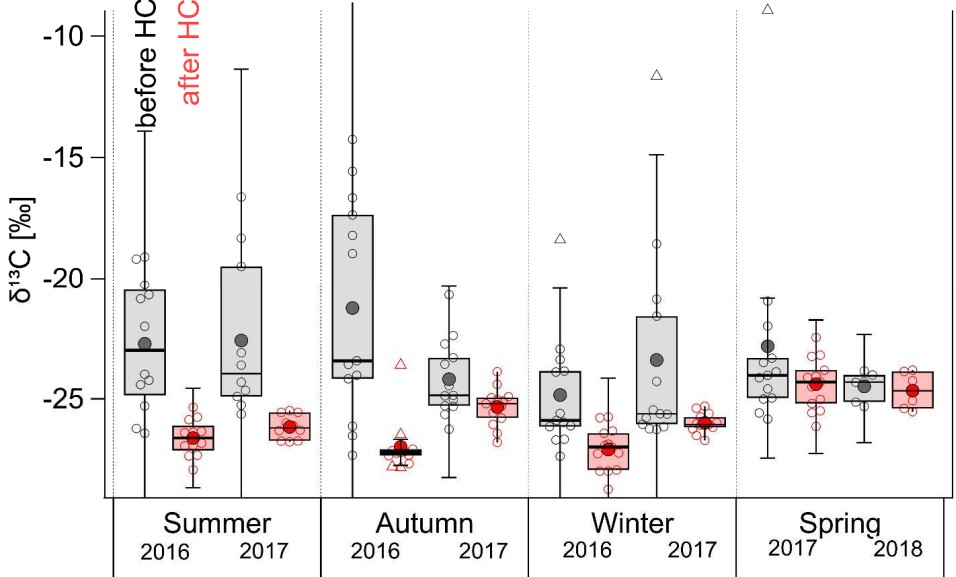


**Figure 6.** Seasonal variations of δ¹³C of TC of untreated (grey) and HCl treated (red) TSP samples at Alert
site from summer 2016 to spring 2018. The boxes correspond to the interquartile range (IQR; 25 and 75
percentile) with median represented by the inner solid line. The whiskers correspond to inner fences range
(1.5*IQR), triangles are outliers and mean is represented by large filled circle.

In addition, measurements of δ¹³C in OC, pyrolytic carbon (POC) + CC, and EC, performed at Environment
Canada Toronto Laboratory, using  ECT9 temperature protocol (Huang et al., 2006, 2021) for the fine
particle (PM₁) samples collected around 2003,  support that a noticeable fraction of CC occurs during the



summer months, as indicated by the relatively more positive $\delta^{13}C$ of "POC+CC" fraction (**Fig. S8**).
Carbonate from eroded rocks in terrestrial environment usually generates large participles, so the CC
content in $PM_1$ should be lower than that in TSP. Consequently, the $\delta^{13}C$ TC in $PM_1$ is expected to be
relatively less positive of compared to that that in TSP. Therefore, **Fig. S8** suggests that the impact of CC
on $\delta^{13}C$ TC is an annual phenomenon occurring over decadal periods. The tendency towards relatively more
positive $\delta^{13}C$ in summer OC fraction also suggests the presence of minor salt oxalate, or potassium or
magnesium carbonates, which are released at 550 °C or lower, as shown in **Fig. S2**. In the time series in
**Fig. 7**, we observed episodes with significant differences in $\delta^{13}C$ values and alternating over short periods.
The comparison of $\delta^{13}C$ of TC and TC mass concentration for both HCl untreated and treated samples
shows insignificant correlation (**Fig. S9**). However, we found a significant, though not strong, correlation
of $\delta^{13}C$ with wind speed (**Fig. S10**) (r = 0.36) and a negative correlation after HCl fumigation (r = -0.32 for
$\delta^{13}C_{TC-HCl}$ vs. wind speed). Concentrations of CC were also positively correlated with wind speed (r = 0.48,
**Fig. S10**), indicating that wind has an effect on this aerosol component. Therefore, we compared the $\delta^{13}C$
time series with average wind speeds and prevailing AMBTs from the HYSPLIT model (see colored bars
in **Fig. 7**), which were divided into six regions as shown in **Fig. 1.**
The large differences in $\delta^{13}C$ values (or relatively more positive $\delta^{13}C_{TC}$ values) can be divided into three
categories. The most frequent differences were observed during periods of stronger winds (average WS
above 4 m s$^{-1}$) associated with the prevailing back trajectories from the North Canada region. These episodes
can be found mainly in summer and autumn 2016 and summer 2017. Such conditions favor dust
resuspension, which may also contain limestone, known to be abundant in this region (Not and Hillaire-
Marcel, 2012; Phillips and Grantz, 2001). The presence of a peak in soil dust carbonates in late
summer/early autumn  is consistent with multidecadal aerosol aluminum observations at Alert (Sharma et
al., 2019; Sirois and Barrie, 1999). These observations also indicate a peak in soil dust aerosol in the spring
month of May (Sharma et al. 2019).
Second, in late February/March 2018, we observed significant enrichment of $^{13}C$, which may probably be
linked to long range transport (LRT) from/over Europe, Greenland and North America (Fig. 6). Sources of
carbonate in this case may be, for example, calcifying marine phytoplankton (Monteiro et al., 2016), which
are abundant in the North Atlantic (Okada and Honjo, 1973). Another possibility is the volcanic and
subarctic semi-desert areas in Iceland (Arnalds et al., 2016).

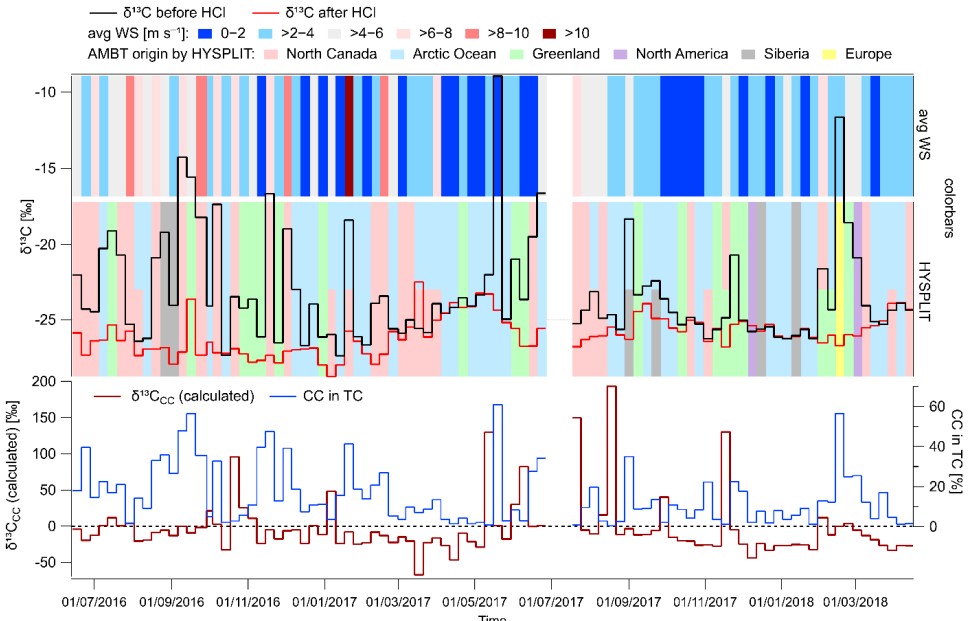

**Figure 7.** Time series of isotopic composition ($\delta^{13}$C) of TC before and after HCl treatment of samples (upper part) with color bars representing average wind speed (top) and origin of AMBT based on HYSPLIT model and divided to regions shown in Fig. 1 (middle part). Time series of the calculated $\delta^{13}C_{CC}$ together with contribution of CC in TC (bottom).

The third case is a $\delta^{13}$C difference observed during lower wind speeds coming from the Arctic Ocean or Greenland. This includes also the sample taken between 15-22 May 2017, whose thermograms are shown in **Fig. 3**, and which was most enriched in $^{13}$C carbon. A recent study by Gu et al. (2023), which reports observations of summer carbonaceous aerosols in the Arctic Ocean, is relevant in this context. They also observed relatively more negative $\delta^{13}$C values of TC, but in this case they did not consider the enriched $^{13}$C carbon as a carbonate contribution, instead, they associated it with an input of fresh marine particulate organic carbon (MPOC) (Verwega et al., 2021). MPOC is formed by the conversion of inorganic carbon by marine phytoplankton through photosynthesis in ocean surface layer (Descolas-Gros and Fontugne, 1990), and this carbon can be partially released into the atmosphere as marine aerosol (Ceburnis et al., 2016). We cannot exclude the influence of MPOC on the TSP taken at the Alert station, but the specific EC/OC thermogram (**Fig. 3, Fig. S2**) shows rather an influence of CaCO$_3$. The presence of carbonates in surface seawater and their interference with organic coatings has been known for decades (Chave, 1965).



Dissolved $CO_2$ in the oceans consists mainly of inorganic substances, which are bicarbonates ($HCO_3^-$;
>85%) and carbonates (ca 10%), and their content varies with temperature, pH, salinity and other
parameters (Zeebe, 2012). Carbonate, in the form of $CaCO_3$, is generally supersaturated in surface seawater
but its precipitation may be limited due to dissolved organic matter (Chave and Suess, 1970). In addition
to common inorganic reactions due to dissolved $CO_2$, the carbonate cycle is also influenced by marine life.
Phytoplankton such as coccolithophores (e.g. *Emiliania huxleyi*), could also contribute to formation of
carbonates (Smith et al., 2012). These organisms produce calcified shells called coccoliths, which are about
2–25 μm across (Monteiro et al., 2016). While these microfossils are mostly deposited on the seabed, they
are also likely to be released also into the atmosphere with marine aerosol from the upper sea layers.
Whether through MPOC or inorganic carbon, this phytoplankton influences the fractionation of $^{13}C$ carbon
(Holtz et al., 2017).
Coccolith microfossils contain enriched calcite with $\delta^{13}C$ values around 0 ‰ (McClelland et al., 2017).
Limestone sediments in the Canadian Arctic are even more enriched, with $\delta^{13}C$ values ranging from 2 to
8 ‰ (Beauchamp et al., 1987). After estimating $\delta^{13}C_{CC}$ (see subsection 2.4), we observed that with a high
CC abundance in TC, the calculated $\delta^{13}C_{CC}$ values are also seen around 0 ‰ (**Fig. 7** bottom, **Fig. S5**). This
suggests that when the TC contains a larger contribution of CC (suppose above 20%), it can be assumed
that a significant portion of the nominal CC is derived from limestone. However, the uncertainties in the
determining of $\delta^{13}C_{CC}$, mentioned in section 2.4, primarily due to the low CC contributions in TC, likely
prevent distinguishing whether the carbonates are from sediment resuspension or marine calcified shells.
Some insight, however, can be obtained from the AMBT analysis discussed earlier.
Finally, **Fig. 8a** confirms a strong dependence of $\delta^{13}C_{TC}$ on the CC content in TC (r = 0.79). The dependence
of $\delta^{13}C_{TC}$ directly on CC mass concentration is even stronger (r = 0.85, after excluding 1 outlier, **Fig. S11**).
Furthermore, we calculated the fractions of CC for individual samples via isotope mass balance (using a
$\delta^{13}C$ of CC value of zero as the end member, see subsection 2.5.) and found that, overall, the calculated
results were approximately 5 % lower than the measured CC/TC, with less scatter. In addition, we observed
an excellent correlation between $\delta^{13}C$ of TC and the calculated fraction of CC, with r = 0.96 shown in **Fig.
8b**, indicating that more than 92 % of the variation in $\delta^{13}C$ of TC can be explained by the dependency on
the fraction of CC.



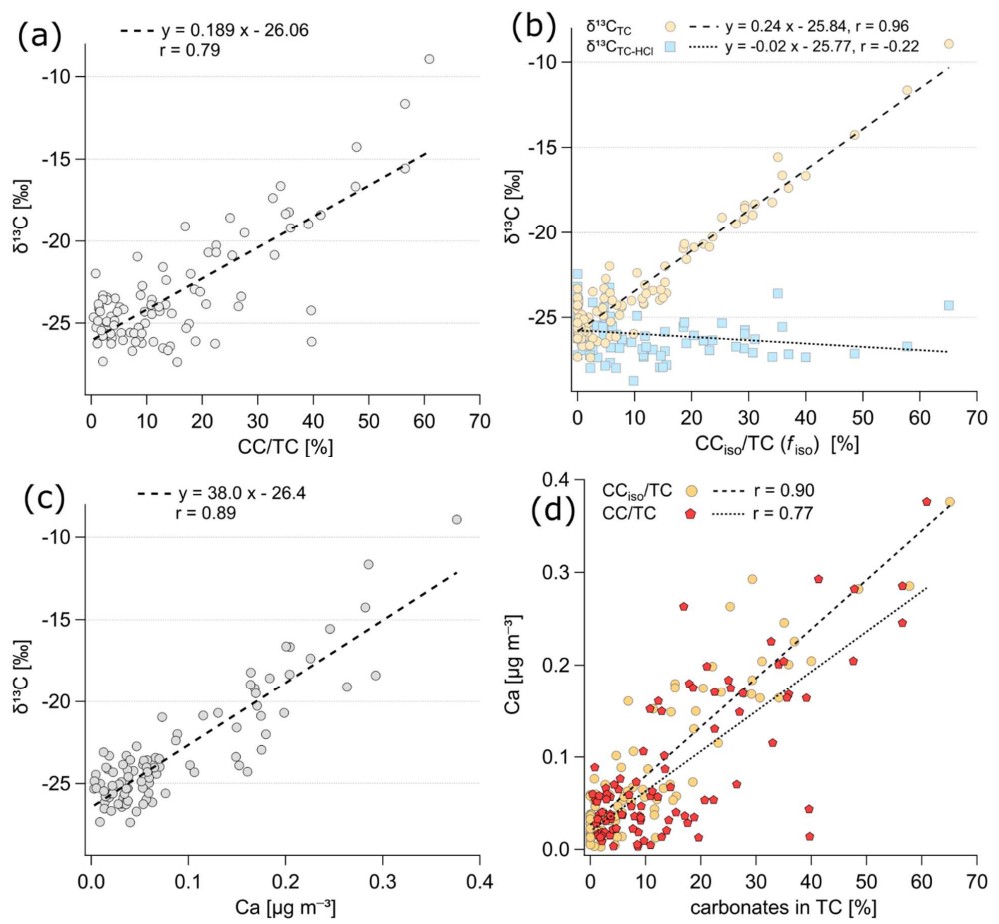


**Figure 8.** Dependence of (a) $\delta^{13}C_{TC}$ on the percentage contribution of CC in TC, (b) $\delta^{13}C_{TC}$ and $\delta^{13}C_{TC-HCl}$
on the calculated fraction of $CC_{iso}$ in TC, (c) $\delta^{13}C_{TC}$ on Ca mass concentrations, and (d) Ca mass
concentrations on percentage contribution of CC and $CC_{iso}$ in TC.


Further support for the influence of $\delta^{13}C$ of TC in favor of carbonates is provided by its strong correlation
with calcium (r = 0.89, **Fig. 8c**). Calcium is also strongly correlated with CC contribution in TC, with r =
0.90 for Ca vs. $CC_{iso}$/TC (**Fig. 8d**). Apportionment of Ca amongst aerosol sources at Alert site (Sharma et
al., 2019), using multidecadal observations and Positive Matrix Factorization analyses, showed that, on



average for both the period November to February, and March to May, Ca was associated 84 and 85% with
windblown dust/soil and 12 and 8 % with sea salt aerosol, respectively.
We also investigated the possible contribution of carbon from magnesium carbonate. Magnesium was
strongly correlated with sodium (r = 0.91, **Fig. S12**), indicating its link mainly to marine aerosol. In contrast,
we observed no relationship between Mg and Ca; this dependence was strongly scattered (**Fig. S12**). Overall,
the results indicate that the main contribution of CC, that strongly influences the $\delta^{13}C$ of TC, is mainly
$CaCO_3$. If there is a contribution of magnesium carbonate, it is rather episodic.
These results thus provide evidence that the CC content in aerosols, mostly of soil origin and to a lesser
extent marine origin, strongly influences the $\delta^{13}C$ isotopic composition of TC in the Arctic atmosphere.
Further research at different Arctic sites could reveal whether the non-negligible presence of CC in the TSP
is the case only in the northern Canada region or a phenomenon observe in larger parts of Arctic.

**4 Summary and conclusions**

We found that the aerosol CC (i.e. carbonate carbon) fraction in Arctic TSP at Alert site is not negligible.
The relative abundances of CC in TC ranged from 0 to 60 % with an average of 11 % over the entire
measurement period. On average, 25 % (range: 0 to 81 %) of EC and 12% (range: 0 to 46 %) of OC was
identified here as nominal CC. The influence of CC removal from the sample was also significantly
reflected in the isotopic composition of $\delta^{13}C$ of TC. The effect of CC on $\delta^{13}C$ was particularly pronounced
in the summer of 2016 and 2017, as well as during autumn 2016 due to strong local Arctic dust transport.
In contrast, the effect of removing of CC on $\delta^{13}C$ was lowest in spring. Thus, CC content in TSP at Alert
can sometimes strongly influence the $\delta^{13}C$ values of aerosols.

Based on the thermograms from EC/OC analyses and the calculated $\delta^{13}C_{CC}$, whose values were around 0‰
at high CC contributions to TC, we conclude that the major part of CC is derived from carbonates.
Additionally, based on the isotope mass balance calculation (using 0‰ as $\delta^{13}C_{CC}$), an excellent dependency
between $\delta^{13}C$ of TC and the calculated fraction of CC (r = 0.96) is observed, supporting that most of the
variation of $\delta^{13}C$ of TC were due to the contribution of CC. Based on the AMBT analyses, we identified
two possible carbonate sources. The first is eroded and resuspended limestone sediments in the northern
Canadian region. The second source may be calcareous shells (coccoliths) produced by marine
phytoplankton and transported from both the Arctic Ocean and the North Atlantic Ocean. However, the
hypothesis of these sources requires further detailed research.



In general, when analysing $\delta^{13}$C of TC in coarse aerosol or aerosols laden with dust, it must be taken into
account that the resulting values may be strongly influenced by CC content. If CC is not removed prior to
EC/OC analysis, CC may be mistakenly identified as EC during Improve_A analysis. This could, for
example, affect modelling of the effect of carbonaceous aerosols on the Arctic climate, as EC (or black
carbon) has a warming effect on the atmosphere, while CC likely has the opposite effect.

**Data availability.** All relevant data for this paper are archived and are available upon request from the
corresponding authors or online at repository here: https://zenodo.org/records/14204515

**Supplementary data**
Supplementary data to this article can be found as a pdf file uploaded together with this manuscript.

**Author contribution.** All authors contributed to the final version of this article. PV analyzed EC/OC
before and after HCl treatment, as well as $\delta^{13}$C of TC, AMBT analyses, evaluated all data and wrote the
paper under the supervision of KK. BK analyzed $\delta^{13}$C of TC after HCl treatment as well as supporting water
soluble ions measurements. LH calculated contribution of $CC_{iso}$/TC based on $\delta^{13}$C measurements. DD and
MMH assisted in the gravimetry and other sample treatments. SS provided meteorological data. SS with
KK and LB managed the field campaign. All authors provided advice and feedback throughout the drafting
and submission process.

**Competing interests.** Kimitaka Kawamura is an editor for Atmospheric Chemistry and Physics. The
authors declare that they have no other conflict of interest.

**Acknowledgements.** This study was supported by JSPS grant no. 24221001, the JSPS Joint Research Program
implemented in association with DFG (JRPs-LEAD with DFG: JPJSJRP 20181601) and from the Ministry of
Education, Youth and Sports of the Czech Republic under project ACTRIS-CZ LM2023030. Authors thank the CFS
Alert for maintaining the base, Andrew Platt Alert coordinator, the operator for ECCC and students for sample
collection at Alert and shipment of these samples.





**Table 1:** Seasonal averages ± standard deviations (medians in parentheses) of different variables in TSP at
Alert site.

| | Summer 2016 | Autumn 2016 | Winter 2016 | Spring 2017 | Summer 2017 | Autumn 2017 | Winter 2017 | Spring 2018 |
|---|---|---|---|---|---|---|---|---|
| N | 12 | 13 | 13 | 13 | 10 | 13 | 13 | 6 |
| OC [µg m$^{-3}$] | 0.182±0.075 (0.177) | 0.136±0.072 (0.106) | 0.253±0.171 (0.237) | 0.184±0.092 (0.151) | 0.179±0.071 (0.155) | 0.106±0.06 (0.128) | 0.154±0.057 (0.177) | 0.154±0.053 (0.153) |
| EC [µg m$^{-3}$] | 0.037±0.021 (0.032) | 0.056±0.055 (0.023) | 0.100±0.121 (0.070) | 0.057±0.037 (0.047) | 0.031±0.015 (0.026) | 0.019±0.013 (0.017) | 0.055±0.036 (0.046) | 0.032±0.018 (0.029) |
| TC [µg m$^{-3}$] | 0.219±0.082 (0.216) | 0.193±0.120 (0.126) | 0.353±0.287 (0.290) | 0.240±0.113 (0.186) | 0.210±0.076 (0.186) | 0.125±0.070 (0.146) | 0.209±0.088 (0.225) | 0.186±0.071 (0.182) |
| OC$_{HCl}$ [µg m$^{-3}$] | 0.150±0.067 (0.144) | 0.103±0.052 (0.095) | 0.215±0.125 (0.201) | 0.172±0.088 (0.149) | 0.159±0.062 (0.150) | 0.096±0.051 (0.111) | 0.140±0.052 (0.161) | 0.146±0.056 (0.138) |
| EC$_{HCl}$ [µg m$^{-3}$] | 0.022±0.008 (0.022) | 0.024±0.012 (0.021) | 0.067±0.052 (0.058) | 0.040±0.015 (0.040) | 0.019±0.004 (0.019) | 0.014±0.007 (0.015) | 0.034±0.018 (0.031) | 0.029±0.018 (0.024) |
| TC$_{HCl}$ [µg m$^{-3}$] | 0.172±0.070 (0.165) | 0.127±0.061 (0.115) | 0.282±0.175 (0.250) | 0.211±0.101 (0.175) | 0.178±0.064 (0.169) | 0.110±0.057 (0.128) | 0.174±0.068 (0.194) | 0.175±0.074 (0.162) |
| CC [µg m$^{-3}$] | 0.047±0.028 (0.052) | 0.066±0.068 (0.030) | 0.071±0.131 (0.034) | 0.029±0.051 (0.010) | 0.032±0.036 (0.020) | 0.015±0.018 (0.006) | 0.035±0.048 (0.014) | 0.011±0.011 (0.007) |
| TC/mass [%] | 10.3±3.8 (9.4) | 6.0±1.9 (5.9) | 7.8±3.5 (8.3) | 5.9±2.4 (5.4) | 13.2±5.4 (13.7) | 8.3±6.2 (7.3) | 6.0±1.6 (5.2) | 7.8±1.8 (7.3) |
| CC/TC [%] | 21.1±11.2 (19.5) | 26.1±19.8 (32.7) | 15.2±10.6 (13.8) | 9.6±15.9 (4.3) | 13.5±14.2 (6.2) | 10.2±7.1 (8.9) | 13.2±15.2 (8.0) | 6.8±6.4 (4.4) |
| CC$_{iso}$/TC [%] | 15.0±9.6 (14.9) | 22.0±16.0 (15.6) | 8.5±8.5 (6.7) | 7.8±17.9 (1.2) | 14.2±12.6 (9.8) | 4.6±5.3 (4.4) | 10.3±17.4 (0.2) | 1.2±2.4 (0.0) |
| OC$_{HCl}$/TC [%] | 68.3±12.5 (69.1) | 59.0±18.1 (55.5) | 65.6±9.5 (68.9) | 72.5±13.9 (75.7) | 76.5±12.8 (81.8) | 77.7±6.3 (79.0) | 71.0±13.4 (76.4) | 78.5±4.6 (79.5) |
| EC$_{HCl}$/TC [%] | 10.6±3.0 (10.5) | 14.9±8.5 (12.7) | 19.2±2.7 (20.2) | 17.9±5.1 (16.5) | 10.0±3.6 (9.3) | 12.0±3.2 (11.6) | 15.8±4.7 (15.8) | 14.7±3.4 (13.3) |
| TC$_{EA}$ [µg m$^{-3}$] | 0.218±0.101 (0.214) | 0.184±0.118 (0.120) | 0.345±0.263 (0.287) | 0.234±0.119 (0.200) | 0.199±0.079 (0.174) | 0.126±0.067 (0.124) | 0.227±0.100 (0.224) | 0.195±0.083 (0.191) |
| δ$^{13}$C$_{TC}$ [‰] | -22.7±2.7 (-23.0) | -21.2±4.5 (-23.4) | -24.8±2.4 (-25.9) | -22.8±4.4 (-24.0) | -22.6±3.2 (-24.0) | -24.2±1.6 (-24.9) | -23.4±4.3 (-25.6) | -24.5±0.6 (-24.3) |
| TC$_{EA-HCl}$ [µg m$^{-3}$] | 0.141±0.063 (0.136) | 0.092±0.052 (0.079) | 0.273±0.196 (0.253) | 0.199±0.108 (0.169) | 0.157±0.065 (0.144) | 0.103±0.052 (0.099) | 0.173±0.068 (0.188) | 0.184±0.080 (0.166) |
| δ$^{13}$C$_{TC-HCl}$ [‰] | -26.6±0.7 (-26.6) | -27.0±1.1 (-27.2) | -27.1±0.9 (-27.0) | -24.4±1.1 (-24.3) | -26.2±0.5 (-26.2) | -25.3±0.8 (-25.2) | -26.0±0.4 (-26.1) | -24.7±0.8 (-24.7) |
| WS [m s$^{-1}$] | 5.1±1.8 (4.5) | 5.5±2.7 (5.6) | 3.9±2.7 (3.1) | 3.0±1.4 (3.5) | 4.2±1.8 (4.6) | 2.5±1.2 (2.1) | 3.6±1.8 (3.4) | 2.7±0.6 (2.6) |
| Temp. [°C] | 5.0±4.6 (4.2) | -13.8±6.9 (-14.0) | -28.7±4.0 (-27.6) | -20.5±8.4 (-21.3) | 1.2±4.6 (1.5) | -18.6±6.8 (-19.6) | -26.2±3.9 (-25.5) | -30.0±3.5 (-30.8) |




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
