# Peer review of "Carbonate content and stable isotopic composition of"

_EGUsphere, 2024_

## Author Response (AR1)

Dear editor,

Thank you for processing and reviewing our manuscript. We are sending responses to both reviewers' comments, including changes in manuscript. Comments are in black, our answers are in blue.

At the same time, we would like to draw your attention to a change in the author team. Ambarish Pohkrel has been added - he was involved in the evaluation of ion chromatography data and we forgot to list him in the previous version. Furthermore, author "Dhananjay K. Deshmukh" has requested to shorten his name to "Dhananjay Kumar" only. The other co-authors agree with these changes.

**RC1**: 'Comment on egusphere-2024-3656', Anonymous Referee #1, 05 Mar 2025 reply

Comments to Carbonate content and stable isotopic composition of aerosol carbon in the Canadian high arctic

I am a scientist who specialize more in the area of radiocarbon instrumentation and the analytical method.

Thank you for reviewing of submitted manuscript. Responses to individual comments and their addressing is provided separately below.

Introduction

The claim of line 62-63 is too strong or even false about the influence of carbonates on the atmosphere in the Arctic of not been studied yet. Just go to Google scholar and type arctic carbonate aerosol and you will get several publications.

Based on this comment, we reviewed the literature again. There is a large number of publications on the topic of "Arctic aerosols" or "carbonates in the Arctic". Articles on carbonates in the Arctic usually deal with their content in the soil or in the sea/ocean water. In terms of atmospheric aerosols and carbonates, we have added the following relevant study - (Mukherjee et al., 2020). In this context, we have added following new sentence to introduction: "In addition to dust, it is hypothesized that a source of carbonates in summer Arctic aerosols may be marine microorganisms from sea spray as reported by Mukherjee et al. (2020) based on calcium analyses."

However, we could not find a study that systematically address a carbonate content in Arctic atmospheric aerosols for more than one year. In contrast, carbonates are not mentioned as a potential aerosol species in terms of their current or future occurrence in the Arctic (Schmale et al., 2021). From this perspective, the significance of this manuscript is novel. If the reviewer has a specific tip for a systematic study on atmospheric aerosol carbonates in the Arctic, we would be happy to receive it.

Finally, to make it clear that this work concerns analyses of atmospheric samples, we made a small change in the manuscript's title as follows: "Carbonate content and stable isotopic composition of atmospheric aerosol carbon in the Canadian High Arctic"

The claim in line 73-74 is misleading. There are many publications that attribute the higher delta13C values to carbonates in the Artic.

Similar to the previous comment, the publications concerning $\delta^{13}C$ content in Arctic carbonates are mostly related to seawater or sediment samples. There is no specific study that addresses changes in $\delta^{13}C$ values related to carbonates in the Arctic atmosphere. In this context, we have modified the sentence in the introduction as follows to make it more clear (red text is newly added): "In the Arctic region, observed higher aerosol $\delta^{13}C$ values in atmospheric aerosols are often attributed to dissolved particulate organic matter from marine aerosol sources while the influence of carbonates is ignored"

There is a conceptual point that the authors do not deal with which is important because this work is about carbonates. If EC is the same as BC but carbonate are colorless thus carbonate is not part of BC then in which group, do we set the carbonates? If the answer is that carbonates are part of EC then this proves that EC is not the same as BC, right? Carbonates belong to what fraction of the carbonaceous matter?

In principle, EC and BC are not the same, as each is measured in a different way and defined differently – see, e.g., Petzold et al. (2013).  In general, carbonates can be classified as inorganic carbon. However, this is different from both EC and OC, for which the EC/OC method of analysis is primarily designed.

Regarding the method, when carbonates are present in a sample, they are either interpreted as OC or EC (depending on the temperature protocol) based on simple thermogram from Sunset analyzer. However, carbonate concentrations are usually negligible (especially in fine aerosol), so they are usually not considered in studies. Nevertheless, when carbonates are present in a sample, they need to be distinguished from OC and EC in a particular method, for which different techniques are used (Jankowski et al., 2008; Karanasiou et al., 2011). We used one of these methods that uses HCl fuming. Here, it is also important to mention that one of the results of this study is a new approach for the estimation of carbonates in aerosol using d13C isotopic composition of TC - see subsection 2.5. The results (Fig. 8) suggest that the proportion of CC in TC determined in this approach is maybe more accurate than by HCl fuming method.

Results

Weak demonstration of quantitative analysis of carbonate with Sunset. I have problems believing that carbonates can be quantitatively analyzed by the proposed Sunset method due to the Sunset lower temperature comparing with elementar analyzer (1000 deg). I do believe that certain amount of carbonate evolves qualitatively during Sunset OC and EC fractions but it is hard to believe that 100% of carbonates becomes CO2 at EC2 temperature.

After seeing Chow et al 2007 and Hu et al 2023, I can not find a method developed and rigorously proved about the quantitatively analysis of carbonate with the Sunset temperature in any of the provided references. Of course, the claim is correct in line 141 "These are temperature steps during which we should expect the release of carbonate carbon". But that does not prove that it is release quantitatively to accurately measure CC as TC-TC$_{HCl}$

Jankowski et al. shows qualitatively that CC evolves at 600-650 deg but the quantitatively part was done measuring TC that includes CC at 1000 deg.

We newly have added references to studies on carbonates analyses in aerosols (Jankowski et al., 2008; Karanasiou et al., 2011). In particular, Karanasiou et al. (2011) discusses the quantification of atmospheric carbonates using the Sunset analyzer and in this respect demonstrates its applicability.

Here we would mention that the EC2 step (740°C) is not the highest used temperatures. In the Improve_A used protocol, step EC3 followed with a temperature of 840°C, but very little mass usually was released during this step. For this reason, we consider the given temperature protocol to be sufficient for the release of commonly occurring carbonates. Another thing is that by using EA method, TC release proceeded at 1000 °C. This may account for the slightly higher TC concentrations from EA (+2%) compared to the EC/OC analyzer (Fig. S1). For this reason, we added the following new sentence at the end of subsection 2.2: "The slightly higher TC concentrations using EA (2 %) may be due to the use of a higher maximum temperature (1000 °C vs 840 °C) for sample release."

The result in figure S1 (left) comparing TC Sunset before any HCl treatment look to me in average lower than the TC from EA-IRMS. The slope (1.014) looks high due to the outlier at approximately 1.1 ug/m3. By removing this outlier with the strongest magnitude, you will get a lower slope and the method from this work (Sunset carbonate) is underestimating, in average. The same applies to figure S4 where the outlier, strongest in magnitude, is lifting up the slope to 0.95.

In Figures S1 and S4, we added a second linear fit that does not include the mentioned outlier. The slope in Fig.S1 changed from 1.01 to 0.93, which is still a good agreement within aerosol analyses. In the case of comparison of CC results using TC analyses from EA and Sunset analyzer in Fig. S4, however, the discrepancy is higher (change of the slope from 0.95 to 0.73). Regarding this, we have made the following change in the text.

"…and the resulting CC values show good agreement (r = 0.87, y = 0.95 x, Fig. S4). Unless otherwise stated, CC values calculated from EC/OC analyses are discussed in this study."

Red is newly added/changed: "and the resulting CC values show acceptable agreement (**Fig. S4**). Fig. S4 shows that CC values based on analyses from the EC/OC analyzer may be slightly underestimated. However, unless otherwise stated, CC values calculated from EC/OC analyses are discussed in this study."

If you check the publication from Baudin et al. 2023 (doi 10.2516/stet/2023038), you can see that the temperature depends on the type of carbonate (Fig. 5, Fig. 6) where they increased all the way until 1200 degrees. Yes, it is a fact that the carbonate evolve temperature decreases with the amount (Fig. 1). Because your manuscript is dealing with very small amounts then you oxidize most of it but conceptually and empirically (Fig. S1 of the manuscript), I believe that your method is losing a small part and it is not quantitative. If true, you should mention this.

The only carbonate released above 900°C in work of Baudin et al. (2023) is strontianite, which is a rare carbonate mineral that we do not expect to be significantly abundant - see, for example, the negligible abundance of strontium in Arctic aerosols (Landsberger et al., 1990).

Anyway, we have taken this particular comment into account and we have added following new sentence to the subsection 2.3.: "This procedure is probably not able to analyze all carbonates

(Baudin et al., 2023) but quantitatively, this method leads to the analysis of at least 90% of the carbonates in samples (Karanasiou et al., 2011).”

Source apportionment

It is hard to believe that the origin of CC is mainly from North America as claimed in the abstract and conclusions. In Fig. 7, it seems that the stronger changes of d13C are due to North America but with episodes coming from Greenland too. For example, in Fig. 7, at 01/03/2018 there is a very strong episode (peak of d13C before HCl) colored in green with yellow (Greenland, Europe). Or a mixture of both, for example at 01/09/2016 the peak of d13C before HCl is a mix of green and pink (Greenland, North America). Later in the text, the authors did explain that the influence is also coming from Greenland, Europe but why the conclusion of North America as main source in the abstract and conclusion?

We mention in both the abstract and the conclusions that, in addition to a main source of carbonates in dust from sites in northern Canada, we have identified other source, which is likely to be marine phytoplankton.

In the abstract, see: “Another potential minor source of CC is -marine aerosol -, including calcified marine phytoplankton shells (coccoliths) introduced into the atmosphere via sea-to-air emission.”

In the conclusions, see: ” The second source may be calcareous shells (coccoliths) produced by marine phytoplankton and transported from both the Arctic Ocean and the North Atlantic Ocean.”

As you correctly note, we discuss this phenomenon in more detail in subsection 3.2 and in Fig. 7. The conclusions that main source of carbonates is probably north Canadian resuspended dust is based on both the location of Alert station and on prevailing winds there (see supplementary data in repository). Both the conclusions and especially in the abstract are usually limited in length, and the findings there are presented briefly. In this respect, we therefore consider the current references to the marine carbonate source in these sections to be sufficient.

RC2: 'Comment on egusphere-2024-3656', Anonymous Referee #2, 22 Mar 2025 reply

The manuscript “Carbonate content and stable isotopic composition of aerosol carbon in the Canadian High Arctic” by Vodička et al., utilizes measurements from the arctic to quantify the contribution of carbonate to total suspended particle mass. The manuscript is clearly written and of interest to the community. My comments are included below

We thank the reviewer for his insightful comments. We have tried to deal with them as best as possible. As part of addressing the comments, we have also performed additional analyses for some standards, as noted in the comments below.

- The authors mention some of the interferences for carbonate quantification, such as oxalates. Is it possible to provide a bound on the uncertainty arising from this. For example, in

the text the authors state that oxalates peaks likely appear in the OC4, while $CaCO_3$ appears in EC2. The mass lost in the EC2 fraction may provide a lower bound for the carbonate fraction.

Determining a bound on the uncertainty in the CC measurement is difficult in this case. The resolution of oxalate in OC4 and $CaCO_3$, follows from both our observations and the literature. However, as can be seen, for example, from Fig. 2a, other types of carbonate may be released in OC4 that are not as common but are counted in the CC. Other compounds in the sample may then also have an effect on the carbonate release temperature (Karanasiou et al., 2011). Some literature mentions possible removal of salts of volatile organic acids together with carbonates by HCl fuming (Chow et al., 1993) which may interfere with CC determination. However, we could not find specific examples with $CaCO_3$ compared to oxalate, before and after HCl fuming. Therefore, we performed additional analyses, whose results are newly included in **Fig. S2b** in supplement. In the case of $CaCO_3$ it appears that after 2h its removal was similar to that after 20h. In the case of potassium oxalate monohydrate, which was released as a representative of the less volatile organics in the OC4 step, there was a surprising increase in the concentration of this peak after HCl fuming. It follows that HCl did not directly lead to the removal of this organic material, but had an effect on its modification, which was reflected in the thermogram. This may also be the case for other organics and their charring during EC/OC analysis as e.g. mentioned by Jankowski et al. (2008). In both cases ($CaCO_3$ and oxalate), artefacts appear in the thermogram, which make an exact quantitative analysis difficult.

We have added the following text in the manuscript regarding this comment (red is newly added): "Nevertheless, it cannot be excluded that some of the prominent CC peaks in the OC4 region are also from oxalates as confirmed also by additional analyses of $CaCO_3$ and potassium oxalate monohydrate standards before and after HCl fumigation (**Fig. S2b**)."

- Line 140: The authors state that for some samples the largest material loss in EC2, while for others it is in EC1 or OC4. Is the presence of the $CaCO_3$ peak in EC2 repeatable, or is there a situation where it is shifted to lower temperatures, like EC1. Asked another way, is the loss of material at these lower temperatures suggestive that compounds other than $CaCO_3$ dominates the material lost during HCl fumigation.

The presence of $CaCO_3$ in the EC2 region is repeatable assuming a stable temperature profile in the analyzer and the same $CaCO_3$ sample. Hypothetically, minor temperature changes can occur, e.g. when changing hardware (heating filaments, glass oven or temperature sensor in the analyzer), which can lead to minor changes in the real temperature profile even when using the same temperature protocol. Another factor is the sample itself. Carbonate is part of the dust mix where it usually does not contribute more than 50%. Depending on the carbonate fraction in the sample and interference with other substances, $CaCO_3$ can be released at lower temperatures as mentioned by Chow et al. (2001). Nevertheless, even if carbonate peaks may shift in the thermogram, removal of carbonate by HCl fumes should be considered as a determining factor in their analysis. From this point of view, the shift of the peak in the thermogram (e.g. between EC1 and EC2) is not essential. From this perspective, it is also not possible to distinguish that the material lost after HCl fumigation is a form of $CaCO_3$, other carbonates (e.g. dolomite) or possibly other material.

- Were any tests of HCl fumigation performed on the carbonate standards or are there references for this procedure to ensure there is quantitative removal of carbonate.

In some works, quantitative methods are focused on carbonate standards analysis but are not tested on aerosol samples (Baudin et al., 2023). In literature on aerosol analyses by EC/OC analyzer, experiments with carbonate standards are mentioned (Cavalli et al., 2010; Karanasiou et al., 2011). Karanasiou et al. (2011) reported carbonate recovery from samples containing 25-250 ugCC/sample at around 90%. It can be assumed that at least similar recovery rates can be achieved with lower amounts of CC corresponding to our sample concentrations. However, as reported elsewhere (Jankowski and Chow 1993), other compounds (e.g., salts of carboxylic acids) may also be transformed during HCl fumigation, which introduces an uncertainty into the quantification of carbonates.

Regarding to this comment, we have added following new sentence to the subsection 2.3.: " This procedure is probably not able to analyze all carbonates (Baudin et al., 2023)  but quantitatively, this method leads to the analysis of at least 90% of the carbonates in samples (Karanasiou et al., 2011)."

- One of the important concluding points in section 4 is that the mis-identification of CC as either EC or OC will result in incorrect estimates of aerosol effects on radiative forcing.  Are there any previous reports on the optical properties of carbonate aerosol to help expand on this point?

A study by Raman et al. (2011) reported a positive radiative effect of carbonate rich dust, although this effect is lower than that of particles from transport, which are assumed to contain more BC. However, this study was conducted in  India, where the composition, and therefore the color, of the dust may be different, potentially leading to a different radiation effect. Similarly, Chen et al. (2021), shows the effect of carbonates and BC on the calculation of the radiative forcing of dust in Tajikistan. However, these are studies where CC is part of dust, which may have regionally different compositions and thus a different influence on the radiative effect. In general, however, determining the effect of Arctic CC and/or dust on radiative forcing is beyond the scope of this study. It would be a broad topic for a future study.

Regarding this comment, we have added following new sentence to introduction: "It is reported that carbonate rich dust also has a slightly positive effect on radiative forcing, although lower than BC (Chen et al., 2021; Raman et al., 2011)."

In conclusions, we have also changed word "opposite" to "lower" as follows: "This could, for example, affect modelling of the effect of carbonaceous aerosols on the Arctic climate, as EC (or black carbon) has a warming effect on the atmosphere, while CC has a lower warming effect."

- Figure 3:  I find this figure hard to interpret, and I recommend simplifying it, or splitting into different panels.  There are a number of items on it that are not discussed.  For example, the laser intensity.  Also, the textboxes "PC",  "HCl" and "EC" and the short dashed lines around 600 seconds are not defined.  There are also a few periods (OC3 and OC4) where the HCl treated is higher than the untreated.  Is there an explanation for this?

Fig. 3 has been simplified. It is true that some data, such as laser intensity, are not relevant in this case to show material loss by HCl fumigation. For this reason, we have removed them and thus we made this graph more readable.

The higher periods in OC3 and OC4 after HCl treatment are probably due to the transformation of some organic substances by HCl fumigation into less volatile ones (e.g. polymers and other substances), which are released at higher temperatures. In our case, however, it is still OC. This phenomenon is documented, e.g., by Jankowski et al. (2008). Regarding to this we added following new text to the subsection 2.3: "Exposing samples to HCl can also affect the transformation of organic matter, which is then released at higher temperatures. Therefore, the thermogram after fumigation may not reflect only the expected CC loss (Jankowski et al., 2008)."

Figure 7: Are the outliers for the calculated $\delta^{13}C_{CC}$ that are >100 ‰ simply due to the low abundance of carbonates during those periods? If so, it may be worthwhile to filter out those points as they are misleading.

Yes, it is likely that the $\delta^{13}C_{CC}$ calculation is affected at low CC concentrations. This is also evident, e.g., from Fig. S5 in the supplementary material, and we mention this possibility at the end of subsection 2.4.

You are right that it is not necessary to show the outliers calculated in this way (probably Fig. S5 is sufficient). For this reason, we have adjusted the y-axis range in Fig. 7 for $\delta^{13}C_{CC}$ to range from -40 to +20. This should make Fig. 7 more straightforward to the relevant values of $\delta^{13}C_{CC}$.

Typographical:

Line 57: This is not a complete sentence

We have expanded the sentence (see the red text below) to make it complete.

"Arctic regions are affected by long range transport of dust and also contributed by dust locally"

Line 280: should this be particles?

Corrected. Thank you.

References:

Baudin, F., Bouton, N., Wattripont, A. and Carrier, X.: Carbonates thermal decomposition kinetics and their implications in using Rock-Eval® analysis for carbonates identification and quantification, Sci. Technol. Energy Transit., 78, doi:10.2516/stet/2023038, 2023.

Cavalli, F., Viana, M., Yttri, K. E., Genberg, J. and Putaud, J.-P.: Toward a standardised thermal-optical protocol for measuring atmospheric organic and elemental carbon: the EUSAAR protocol, Atmos. Meas. Tech., 3(1), 79–89, doi:10.5194/amt-3-79-2010, 2010.

Chen, P., Kang, S., Abdullaev, S. F., Safarov, M. S., Huang, J., Hu, Z., Tripathee, L. and Li, C.: Significant influence of carbonates on determining organic carbon and black carbon: A case study in Tajikistan, central Asia, Environ. Sci. Technol., 55(5), 2839–2846, doi:10.1021/acs.est.0c05876, 2021.

Chow, J. C., Watson, J. G., Pritchett, L. C., Pierson, W. R., Frazier, C. A. and Purcell, R. G.: The dri thermal/optical reflectance carbon analysis system: description, evaluation and applications in U.S. Air quality studies, Atmos. Environ., 27A(8), 1185–1201, 1993.

Chow, J. C., Watson, J. G., Crow, D., Lowenthal, D. H. and Merrifield, T.: Comparison of IMPROVE and NIOSH Carbon Measurements, Aerosol Sci. Technol., 34(1), 23–34, 2001.

Jankowski, N., Schmidl, C., Marr, I. L., Bauer, H. and Puxbaum, H.: Comparison of methods for the quantification of carbonate carbon in atmospheric PM10 aerosol samples, Atmos. Environ., 42(34), 8055–8064, doi:10.1016/j.atmosenv.2008.06.012, 2008.

Karanasiou, A., Diapouli, E., Cavalli, F., Eleftheriadis, K., Viana, M., Alastuey, A., Querol, X. and Reche, C.: On the quantification of atmospheric carbonate carbon by thermal/optical analysis protocols, Atmos. Meas. Tech., 4, 2409–2419, doi:10.5194/amt-4-2409-2011, 2011.

Landsberger, S., Vermette, S. J. and Barrie, L. A.: Multielemental composition of the Arctic aerosol, J. Geophys. Res. Atmos., 95(D4), 3509–3515, doi:10.1029/JD095iD04p03509, 1990.

Mukherjee, P., Reinfelder, J. R. and Gao, Y.: Enrichment of calcium in sea spray aerosol in the Arctic summer atmosphere, Mar. Chem., 227(May 2019), 103898, doi:10.1016/j.marchem.2020.103898, 2020.

Petzold, A., Ogren, J. A., Fiebig, M., Laj, P., Li, S. M., Baltensperger, U., Holzer-Popp, T., Kinne, S., Pappalardo, G., Sugimoto, N., Wehrli, C., Wiedensohler, A. and Zhang, X. Y.: Recommendations for reporting black carbon measurements, Atmos. Chem. Phys., 13(16), 8365–8379, doi:10.5194/acp-13-8365-2013, 2013.

Raman, R. S., Ramachandran, S. and Kedia, S.: A methodology to estimate source-specific aerosol radiative forcing, J. Aerosol Sci., 42(5), 305–320, doi:10.1016/j.jaerosci.2011.01.008, 2011.

Schmale, J., Zieger, P. and Ekman, A. M. L.: Aerosols in current and future Arctic climate, Nat. Clim. Chang., 11(2), 95–105, doi:10.1038/s41558-020-00969-5, 2021.